# Effects of Neuromuscular Training on Motor Competence and Physical Performance in Young Female Volleyball Players

**DOI:** 10.3390/ijerph17051755

**Published:** 2020-03-08

**Authors:** Nebojša Trajković, Špela Bogataj

**Affiliations:** 1Faculty of Sport and Physical Education, University of Novi Sad, Novi Sad 21000, Serbia; trajcevolley83@gmail.com; 2Department of Nephrology, University Medical Centre, Ljubljana 1000, Slovenia; 3Faculty of Sport, University of Ljubljana, Ljubljana 1000, Slovenia

**Keywords:** movement competence, skills, youth, training

## Abstract

Although neuromuscular training (NMT) emphasizes injury prevention training, there is little information about its effects on performance in young athletes. This study aimed to investigate the effects of eight-weeks NMT on motor competence and physical performance in 10- to 12-year-old female volleyball players. Sixty-six participants (mean ± SD; 11.05 ± 0.72 years) were randomized into either the NMT group (NTG; n = 32) or control group (CON; n = 34). Sprint on 10-m, modified T-test, plank, vertical jump, and medicine ball throw tests were used to assess the physical performance. The Körperkoordinationstest für Kinder (KTK) was used to assess the subjects’ motor competence. The NMT was performed twice a week during the first 30 min of each regularly scheduled 90-minute volleyball training. Participants in the CON group attended only their regular volleyball training. A significant group x time interaction was found for Motor Quotient KTK (MQKTK) (*p* < 0.001), KTK lateral jumps (*p* < 0.001), and KTK shifting platforms (*p* < 0.01). There was a significant interaction for modified T-test results (*p* < 0.001) and vertical jump (*p* = 0.04). No change was observed in both groups for plank performance (*p* > 0.05). The NMT promoted significant gains in motor competence and physical performance in youth female volleyball players.

## 1. Introduction

Motor competence reflects the degree of proficient performance in various motor skills and is essential for developing an active and healthy lifestyle [1]. If a child has motor problems and is left untreated, he is likely to transfer them into adulthood [2]. Moreover, low motor competence can lead to risks for a mixture of behavioral, emotional, and social difficulties [3]. Additionally, it also significantly impacts the willingness of participation in physical activity and overall performance in different sports [4]. 

Volleyball is one of the most intense anaerobic sports that include a combination of explosive movements with short recovery periods [5,6]. In volleyball, the physical performance plays an essential role, since the actions in this sport include a variety of changes of direction in sagittal and frontal planes, frequent sprints, and different types of jumps [7,8]. Explosive strength represents the ability of the neuromuscular system to manifest strain as quickly as possible and is a crucial fundamental aspect of successful volleyball performance [9]. Besides the importance of physical performance, volleyball can also be considered a skill-based and complex game, which requires well-developed motor coordination levels [10]. The importance of effective motor behavior optimization and decision-making in different motor games was established recently [11]. Moreover, Pic et al. [12] found that girls and boys show differences in the effectiveness of motor behavior and respond differently when they act within a complex interactive structure. Therefore, well-developed programs and methodologies are essential and can significantly contribute to motor skill learning and improving movement competence [13].

Therefore, different kinds of training models have been tested to improve the performance of volleyball players [14,15,16,17]. However, there is a high degree of inter-individual variation in the development of movement competence during early and middle childhood [18]. Accordingly, exercise and health care professionals agree that we should put the focus on appropriate training models for the youth [19]. The evidence says that a supervised strength training program is effective and safe for children and that they can successfully improve their overall health and strength by participating in such a program [20,21]. The neuromuscular training (NMT) program was identified as a new innovative approach for school children [22]. It aims to include general and specific physical activities to enhance health (e.g., endurance and strength) and skill-related (e.g., balance and agility) physical fitness components with a combination of resistance, balance, dynamic stability, plyometrics, core strength, and agility exercises [22]. Faigenbaum et al. [23] reported that the NMT program showed results in the enhancement of children’s fundamental movement skills and fitness. Faigenbaum et al. [23] demonstrated significant improvements after eight weeks of 15 min NMT before-school physical education, in curl-ups, push-ups, 0.5-mile run, and single-leg hop performance, compared to the control group. While the mentioned research adds valuable meaning to NMT, there is still a lack of NMT research on young volleyball players. In adolescent female volleyball players, the NMT program resulted in the improvement of their neuromuscular control and decrease of anterior cruciate ligament (ACL) injury risk by improving dynamic knee stability. Another study showed that regular participation in NMT enhanced countermovement jumping performance in young volleyball players [24]. Sugimoto et al. [25] demonstrated on a sample of 21 high school women volleyball players that the high compliance to NMT significantly elevated the hip abductors isokinetic strength. However, multiple questions remain regarding the efficacy and utility of NMT as a tool to enhance motor competence and physical performance in young athletes, especially in volleyball. Besides that, there is not a sufficient amount of research on this field on the female population. Females may especially benefit from multicomponent NMT since they usually display decreased baseline levels of power and strength in comparison with the male population [26].

Therefore, our study aimed to determine the effects of eight-weeks NMT engagement on motor competence and selected physical performance in 10- to 12-year-old female volleyball players. It was hypothesized that the NMT group would significantly improve their results in the chosen tests compared to the control group.

## 2. Materials and Methods 

### 2.1. Study Design 

Motor competence and physical performance testing were undertaken at the sports facilities of the Department of Sports at the University of Novi Sad in Serbia. The aim was to test the effects of 10-weeks neuromuscular training on motor competence tests, 10-m sprint, change of direction speed (modified T-test), medicine ball throw, and jumping ability (countermovement jump). Pretests and posttests were conducted on two consecutive days, before and after the 10-week intervention period. The study began in February 2019 and was completed in April 2019. Participants from the NMT group underwent a one-week familiarization period before starting the training intervention. The study was conducted during the last two weeks of the precompetition phase.

### 2.2. Participants

A total of 66 youth female volleyball players (age, 11.05 ± 0.72 years; height 157.81 ± 9.18 cm; weight 50.68 ± 11.47 kg) who had been competing in volleyball for two or more years and had no neuromuscular training experience were recruited to participate in the study. Finally, subjects were randomly divided into either the neuromuscular training group (NTG; n = 32) or control group (CON; n = 34). Age, height, and weight data across subjects are presented in Table 1, with no significant differences found between control and experimental groups between the baseline or post-testing sessions (*p* > 0.05). Subjects were selected on the following criteria; currently participating in volleyball training, no prior history of lower limb injuries in the past six months, and no history of resistance training. Written parental consent and subject assent were provided before initiating the study in accordance with the Novi Sad University Human Research Ethics Committee guidelines (ethical approval number: 24/2019).

Analysis of the maturation state [27] revealed that there was no significant difference in maturational categories between the groups (*p* > 0.05).

### 2.3. Testing Procedures

Height was measured to the nearest 0.1 cm with the use of a stadiometer (SECA Instruments Ltd, Hamburg, Germany), while weight was measured to the nearest 0.1 kg on an electronic scale TANITA BC 540 (TANITA Corp., Arlington Heights, IL, USA). Both sessions also included a battery of performance and motor competence tests. 

Sprint on 10 m, T-test for agility, plank, vertical jump, and medicine ball throw tests were used to assess the motor skill performance, and the Körperkoordinationstest für Kinder (KTK) was used to assess the subjects’ motor competence. The order of testing was constant for both testing sessions with the movement competency testing occurring first, followed by the motor skill performance measures. The time of testing for each subject remained consistent for both testing sessions.

### 2.4. Motor Competence Tests

The assessment of the children’s motor competence consisted of four test items of the KTK battery test: walking backward, lateral jumps, one leg jumping, and shifting platforms. The standardization of all test items was demonstrated in previous studies [28,29]. Using normative data tables, based on the performance of a standardization sample, the raw performance score of each test item was converted into a standardized Motor Quotient (MQ) adjusted for age and gender. Adding together all four item MQs resulted in a total MQKTK that was converted into a percentile score and allowed classification in five gross motor coordination levels.

#### 2.4.1. Walking Backward 

The task for a child was to walk backward on three balance beams. The dimensions of the beams were 3 m in length, 5 cm in height, but with decreasing widths of 6, 4.5, and 3 cm. The child had three attempts at each beam, and the number of successful steps was recorded; a maximum of 24 steps (eight per trial) was counted for each balance beam, which comprised a maximum of 72 steps.

#### 2.4.2. Lateral Jumps

The goal of this test for a child was to make consecutive jumps from side to side over a beam (60 × 4 × 2 cm) as quickly as possible in a time window of 15 s. The child was given the instructions to keep his/her feet together. The number of correct jumps in two trials was summed. 

#### 2.4.3. One Leg Jumping 

The child was instructed to hop on one foot at the time, preceding the jump over a stack of foam blocks after a short run-up. After a successful hop with each foot (meaning that the child cleared the block without touching it and successfully continued hopping on the same foot at least two times), the hight was increased by adding new blocks (50 cm, 20 cm, 5 cm). The subject had three attempts at each height and on each foot; three, two, or one point(s) was/were awarded for successful completion on the first, second, or third trial; a maximum of 39 points (12 stacks of blocks) could be awarded for each leg (maximum score of 78).

#### 2.4.4. Shifting Platforms 

The child began the trial by standing with both feet on one platform (25 × 25 × 2 cm; 0.5 kg) supported on four legs, 3.7 cm in height. The subject held the second identical platform in his/her hands; the child was then instructed to place the second platform alongside the first one and then to step onto it; the first platform was then lifted and placed alongside the second, and the child stepped onto it. The sequence lasted for 20 s. Two points were awarded for each successful transfer from one platform to another (one point for shifting the platform, the other for transferring the body onto the second one). The number of points in the 20-s interval was recorded and summed for two trials. If the child fell off the platform in the process, he/she stepped back onto the platform and continued the test.

### 2.5. Physical Performance Tests

#### 2.5.1. 10-m Sprint

The photocells were settled at the starting and finishing points of the distance of 10 meters [30]. All subjects started approximately 0.5 m behind the starting photocell. The best score out of three trials was recorded for further analysis.

#### 2.5.2. Modified T-test

Change of direction speed was evaluated using the modified agility T-test and following the guidelines proposed by Sassi et al. [31]. The subject started behind the photocell (cone A). On the command, the subject sprinted towards the cone B set at the distance of 5 m and touched the base of the cone with the right hand. Then, they turned left and shuffled sideways to cone C (2.5 m), touching the base as well, but this time with the left hand. Then, shuffling sideways to the cone D (5 m), touching its base with the right hand, followed up by shuffle to the left to the cone B (5 m), touching its base with the left hand and running back to the cone A (5 m), finishing photocell. The time was stopped when the finishing photocell was passed by. Trials were deemed unsuccessful if participants failed to touch a designated cone, crossed their legs while shuffling, or were unable to face forward at all times. Subjects performed three trials, and the best score in seconds was taken for further analysis.

#### 2.5.3. Plank

Plank test was modified according to Padulo et al. [32]. This test aimed at holding an elevated position for as long as possible. Starting position was with the upper body supported off the ground by elbows and forearms and the legs straight with the weight taken by the toes. The hip was lifted off the floor, creating a straight line from head to toe. The timing was started as soon as the subject got into the appropriate position. The head faced the ground instead of forward. The test was concluded when the subject was not capable of holding the right position any further. The measured time in seconds to exhaustion was used for further analysis. 

#### 2.5.4. Vertical Jump (Countermovement Jump with Arm Swing)

The vertical jump test was measured using the "OptoJump" system of analysis and measurement [33]. Each jump began in a standing position, motionless. Arm swing was performed with the arms first moving in an extension direction at the shoulder joint, followed by movement in the direction of flexion. The subjects performed three jumps with arm swing and were instructed to jump naturally and as high as they could, performing all jumps with maximal effort. A minimum of 30 s of sufficient rest was allowed between consecutive jumps to prevent the harmful effects of fatigue.

#### 2.5.5. Medicine Ball Throw (Overhead)

The goal of this test was to measure upper body strength and explosive power. Subjects were set to stand on the line with feet side by side but slightly apart and facing the direction to which the ball was thrown [34]. The ball (3 kg) was held with the hands on the side of the ball and slightly behind the center. The ball is brought back behind the head, then thrown vigorously forward as far as possible. The distance in cm from the starting position to where the ball landed during three trials was recorded, and the best one was taken for further analysis.

### 2.6. Neuromuscular Training Program

The NMT program used in this study was designed specifically for volleyball and primary school children, taken and modified from the previous studies [24,35,36]. 

The intervention was performed twice per week (Tuesday and Thursday) during the first 30 min of each regularly scheduled 90-minute volleyball training. The NMT was performed before volleyball practice (after warm-up) to secure total neuromuscular activation. The NTG group had one additional volleyball training for improving volleyball skills and techniques. 

Participants in the CON group did not perform specific NTG but attended only their regular volleyball training three times per week during the study period.

All subjects went through a comprehensive one-week familiarization period in order to ensure familiarity with the types of exercises that were used in the NMT (Table 2). All familiarization and training sessions were initiated with the use of a standardized 10-minute dynamic warm-up (light running, and stretching) followed by plyometric exercises, coordination ladder, strength training, and finishing with the plank. The training was supervised, and players received instructions on how to perform each exercise. Exercises followed a progression, and emphasis was given to the use of a proper technique.

### 2.7. Statistical Analysis

Descriptive statistics were calculated for age, height, and body mass. The distribution of raw data sets was checked using the Kolmogorov–Smirnov test and demonstrated that all data had a normal distribution (*p* > 0.05). Test-retest reliability was assessed for physical performance tests using a one-way intra-class correlation coefficient (ICC) based on average measurements (ICC 1,k). A repeated measure ANOVA (2 × 2) was computed to test for interactions and main effects for time (pretest vs. posttest) and group (NTG vs. CON) on the selected physical test variables. All tests were carried out using SPSS, version 22 (SPSS Inc., Chicago, IL, USA), and assessed at the *p* ≤ 0.05 level of significance. The effect size (ES) of each variable was tested using Cohen’s d within each group and was classified as follows: <0.2 was defined as trivial; 0.2–0.6 was defined as small; 0.6–1.2 was defined as moderate; 1.2–2.0 was defined as large; >2.0 was defined as very large; and >4.0 was defined as extremely large [37], and partial eta (η) squared between groups (0.01 = small effect, 0.06 = medium effect, and 0.14 = large effect) [38].

## 3. Results

All participants completed the study. There were no reports of injury during the study. The ICC for physical performance tests ranged from 0.85 to 0.96.

### 3.1. Motor Competence

Results from repeated measures ANOVA indicated a significant group (NTG vs. CON) × time (Pre to Post) interaction for Motor Quotient Körperkoordinationstest für Kinder (MQKTK) (F 1, 90 = 22.65, *p* = 0.001, Partial ƞ2 = 0.38, See Table 3). Following the NMT, a significant interaction of group and time was found for KTK raw scores, KTK lateral jumps (F = 23.18, *p* < 0.001), and in KTK shifting platforms (F = 9.76, *p* = 0.003). A borderline significant interaction was also found for KTK single-lever jumps (F = 4.1, *p* = 0.05). Large effects favoring the NMT group were observed in KTK lateral jumps (ES = 1.2, % change = 22.2%) and KTK shifting platforms (ES = 1.3, % change = 17.2%), compared to CON group (ES = 0.2, % change = 4.7% and ES = 0.2, % change = 3.1%), respectively.

When examining the impact of the NMT program on KTK walking backward performance, there was a significant main effect for time (*p* < 0.001) with both groups improving their result after the eight-weeks intervention (NTG-ES = 0.6, % change = 15.5% vs. CON-ES = 0.3, % change = 7.4%). There were no significant group x time effects (*p* = 0.065).

### 3.2. Physical Performance

Both NTG and CON demonstrated improvements in sprint10 (4.8% and 4.3%, *p* < 0.001), but no group or interaction effects were observed (*p* > 0.05).

There was a significant group x time interaction when examining the impact of the NTG and CON on the modified T-test results (F 1, 90 = 12.78, *p* = 0.001, Partial ƞ2 = 0.17). The NTG demonstrated a significant decrease in time for the modified T-test (*p* = 0.001, moderate ES), whereas the CON displayed only a small decrease (ES = 0.3, 0.9 % change).

There was no effect of time, group, or interaction on plank performance (*p* > 0.05, Table 4).

There was an interaction (F 1, 90 = 4.42, *p* = 0.04, Partial ƞ2 = 0.07), time and group effect (*p* < 0.05) for vertical jump, with the NTG improving performance by 12.2% (moderate ES) between pre- and post-tests, while a 6.6% (small ES) improvement was observed in the CON group.

When examining the impact of the NMT program on medicine ball throw, there was a significant main effect for the time after the eight-week intervention (*p* = 0.002). There were no significant group x time effects (*p* = 0.4). However, the NTG demonstrated a significant increase on medicine ball throw (*p* = 0.003, ES = 0.5), whereas the CON only displayed a 0.14-m increase (ES = 0.3).

## 4. Discussion

The present study aimed to implement the NMT program into young female volleyball practice and examine its impact on motor competence and physical performance. The main findings of the study were that the eight-week NMT intervention significantly improved MQKTK, KTK single-lever jumps, KTK lateral jumps, and KTK transfer on platforms compared to the control group. Additionally, the NTG improved agility (modified T-test) and vertical jump in young female volleyball players. The largest effect sizes were found for MQKTK (ES = 1.3), KTK lateral jumps (ES = 1.2), and for KTK shifting platforms (ES = 1.3). 

Good motor competence is a significant predictor for high-level sports performance [39], and also for the engagement in sports activities [40]. Additionally, the stability of motor competence levels in childhood may have implications for talent identification purposes as well as potential health-related benefits throughout the lifespan [41]. Deprez et al. [42] showed that results in the KTK test were a good predictor of future dropout and adherence to an elite talent development program in 8- to 16-year-old soccer players. Moreover, Pion et al. [10] showed that motor coordination is an essential factor in determining inclusion into the elite level in female volleyball. Therefore, improving motor competence in young volleyball players could be of great importance. In the present study, we demonstrated that neuromuscular training improved KTK results, and to the best of our knowledge, until now, no published articles have examined the effects of this type of training on motor competence of youth female volleyball players. For NTG, there was seen an improvement in MQ, such as reaching a typical score of 112.3 for age and gender (normal MQ = score 86–115) with a large effect size (ES = 1.3). Similar large improvements were seen in seven–nine-year-old overweight/obese boys following the plyometric training only (large ES, d = 1.02) [28]. Despite the expected positive effects that NMT may offer in terms of children’s development, there are only two published studies regarding motor competence in children [43,44]. Although the studies mentioned above examined the effect of NMT on process-oriented movement skills in slightly younger participants than that investigated in the present study, their data identified positive improvements in their studied variables compared to a control group. Participation in the NMT program resulted in the most significant gains in the KTK lateral jumps (large ES, 22.2%) and KTK transfer on platforms (large ES, 17.2%). These improvements were expected as the NMT program mostly included jumps and hops. Moreover, short shuffling moves in volleyball training, which involve leg coordination [45], could significantly contribute to better results in KTK transfer on platforms. NMT represents an integrative model that includes multiple components (e.g., strength, balance, power, skill development) [46]. However, our program (Table 2) did not include balance exercises, which could be the reason for nonsignificant effects in KTK walking backward test. 

Participants in the NMT group made significantly greater gains in selected physical performance measures following the training period than the CON group. Plyometric training was recognized as the superior training method in improving sprint performance in pre- and mid-pubertal males, where a combined training method was most effective in post-pubertal males [47]. Following puberty, males experience natural increases in strength, muscular power, and coordination that are not generally seen in females [35]. Moreover, Pic et al. [48] revealed motor asymmetry in relation to gender in this age group, which was the expression of behaviors lacking in playful neutrality. Additionally, a general change in girls posture with maturation causes greater knee valgus and lower neuromuscular control compared to boys [49], which may suggest the need for integration of neuromuscular training programs, including plyometrics and strength training, to improve performance. 

Following the intervention, both the NTG (+2.68 cm) and the CON group (+1.31 cm) improved their vertical jump performance; however, the NTG made a larger and significantly greater improvement. This improvement is in agreement with the study conducted by Hopper et al. [36], who reported that 11–13-year-old female netball athletes were able to improve their vertical jump by 4 cm in response to a 6-week NMT program. Also, Chappell et al. [50] found a 4-cm improvement in vertical jump following an NMT program in college female athletes. Both programs were compounded with a combination of balance, plyometrics, and resistance exercises. The possible explanation for the improvement in the CON group is that they were participating in regular volleyball training throughout the study, which requires jumping performance, and this may have contributed to the vertical jump improvement in the CON group. 

The 8-week intervention improved NTG agility measured by the modified T-test by 3%. We demonstrated a significant group x time interaction (*p* < 0.001). Similarly, Hopper et al. [36] examined the impact of a 6-week NMT program on agility determined by 505 Agility Test and found a significant group x time interaction (*p* < 0.001).

In 10-m sprint, plank, and medicine ball throw, we did not show any significant differences compared to the CON group following an 8-week NMT intervention. This may be explained by the fact that our program was more focused on jumps. However, on the contrary, a 12-week plyometric training study [51] with or without a weighted vest in 14–16-year-old female volleyball players investigating the effects on sprint performance showed significant improvements in 50-m sprint test in both groups (with a vest: 1.05 s, no vest: 0.53 s). Another study investigated the effect of six-weeks NMT on 20-m sprint in court-sport athletes and found a significant improvement (2.4% decrease) [36]. On the other hand, a considerable improvement of initial acceleration (10-m sprint) was reported for early-pubertal soccer players after the implementation of NMT in the form of plyometric training [52]. In this study, NMT induced a 4.8% improvement in 10-m sprint. These improvements in speed may be related to increased neuromuscular activation, improved ground contact time, and muscle-tendon unit stiffness [53].

Performance gains in the abdominal plank test following NMT were trivial since the training intervention included only one exercise specifically designed to enhance core strength. Similarly, Faigenbaum et al. [23] found only particularly notable gains in the abdominal curl-up test following NMT in eight-year-old children. 

One of the limitations was that we did not incorporate any specific balance training as part of the NMT program. Moreover, we did not know the participants’ previous involvement in other sports activities. Additionally, because of the short program duration, we do not know what long-term training adaptations would be. Therefore, future studies should be conducted in order to see the long-term adaptations but also whether the program is effective in injury prevention. A vital strength of the present study was the fact that this was the first study that reported improvement in motor competence in young female volleyball players. This improvement in motor competence is of great importance because once these fundamental skills are developed, young female players can begin to participate in strength and conditioning programs. Neuromuscular training seems to be the best way since it sensibly progressed and design to players’ individual needs, goals, and abilities [22].

## 5. Conclusions

The findings of this study indicate that neuromuscular training promotes significant gains in motor competence and physical performance in female youth volleyball players. Our results support the fact that youth in sport should first build an important foundation by developing movement competence, and then build and enrich sport-specific skill sets. This study offers an empirically tested training program that coaches can directly use in practice with female youth volleyball players, and that requires limited equipment and can be appropriate for youth with adequate guidance and instruction from qualified professionals. The results of this study highlight the potential of using neuromuscular training along with volleyball training as a useful, time-efficient, and practical mode of exercise to improve movement competence along with physical performance. Moreover, this study supports the findings that talent identification programs in female volleyball should include motor competence testing to assure a broader performance screening.

## Figures and Tables

**Table 1 ijerph-17-01755-t001:** Age, height, and weight data across subjects.

Group	Age (years)	Height (cm)	Mass (kg)
NTG	11.12 ± 0.68	158.28 ± 8.07	47.83 ± 8.97
CON	10.96 ± 0.75	157.37 ± 10.21	48.59 ± 13.46

Note: results are presented as mean ± SD. Abbreviations: NTG, neuromuscular training group; CON, control group.

**Table 2 ijerph-17-01755-t002:** Neuromuscular training program structure.

Tuesday	Duration	Thursday	Duration
Deep squat	2 sets of 10–12 rep	Wall sits	2 sets, 35–50 s
Forward lunges with Med Ball	2 sets of 7–10 rep	Backward lunges with Med Ball	2 sets of 7–10 rep
Push-ups	3 sets of 7–12 rep	Volleyball push-ups	3 sets of 7–12 rep
Med ball throw (from the chest, side throw)	2 sets of 5–15 rep	Med ball throw (overhead, backward)	2 sets of 5–15 rep
Plank	60-90 s	Crunches, Side plank	60–90 s
Hurdle jumps	2 sets of 8–12 rep	Knee tucks	2 sets of 8–12 rep
Squat jump	2 sets of 6–10 rep	Box jumps	2 sets of 6–10 rep
Four-way jumps (forward, backward, side to side)	60 s	Single leg four-way jumps	60 s
Lunge jumps	8–12 rep	Block jumps	2 sets of 8–12 rep
Agility ladder: Various patterns	2 rep	Agility ladder: various patterns	2 rep

Abbreviations: rep—repetitions; med—medicine; s—seconds.

**Table 3 ijerph-17-01755-t003:** Pretest and posttest results for motor competence.

Group		Pretest	Posttest	ES	% change	*p*-value, η^2^_p_
**MQKTK**						
	NTG	98.58 ± 8.82	112.33 ± 12.36	1.3	13.9 %	Group: *p* = 0.014, η^2^_p_: 0.15 Time: *p* ≤ 0.001, η^2^_p_: 0.66 Interaction: *p* < 0.001, η^2^_p_: 0.38
	CON	95.89 ± 8.60	99.72 ± 8.39	0.5	4.0 %	
**KTK walking backward**						
	NTG	38.81 ± 10.21	44.81 ± 10.77	0.6	15.5 %	Group: *p* = 0.037, η^2^_p_: 0.11 Time: *p* < 0.001, η^2^_p_: 0.36 Interaction: *p* = 0.065, η^2^_p_: 0.09
	CON	33.89 ± 9.66	36.39 ± 9.13	0.3	7.4 %	
**KTK one leg jumping**						
	NTG	37.52 ± 8.87	43.38 ± 11.49	0.6	15.6 %	Group: *p* = 0.031, η^2^_p_: 0.12 Time: *p* < 0.001, η^2^_p_: 0.4 Interaction: *p* = 0.05, η^2^_p_: 0.1
	CON	32.61 ± 8.81	35.06 ± 8.56	0.3	7.5 %	
**KTK lateral jumps**						
	NTG	45.57 ± 6.56	55.67 ± 9.53	1.2	22.2 %	Group: p = 0.52, η^2^_p_: 0.011 Time: *p* < 0.001, η^2^_p_: 0.61 Interaction: *p* < 0.001, η^2^_p_: 0.39
	CON	47.67 ± 9.64	49.89 ± 10.98	0.2	4.7 %	
**KTK shifting platforms**						
	NTG	33.71 ± 4.71	39.52 ± 3.96	1.3	17.2 %	Group: *p* = 0.101, η^2^_p_: 0.07 Time: *p* < 0.001, η^2^_p_: 0.36 Interaction: *p* = 0.003, η^2^_p_: 0.21
	CON	33.94 ± 5.07	35.00 ± 4.79	0.2	3.1 %	

Note: pretest and posttest results are presented as mean ± SD. Abbreviations: NTG, neuromuscular training group; CON, control group; MQKTK, Motor Quotient Körperkoordinationstest für Kinder; KTK, Körperkoordinationstest für Kinder.

**Table 4 ijerph-17-01755-t004:** Pretest and posttest results for physical performance.

Group		Pretest	Posttest	ES	% change	*p*-value, η^2^_p_
**Sprint10** (s)						
	NTG	2.29 ± 0.19	2.18 ± 0.15	−0.6	−4.8 %	Group: *p* = 0.44, η^2^_p_: 0.01 Time: *p* = 0.001, η^2^_p_: 0.16 Interaction: *p* = 0.8, η^2^_p_: 0.001
	CON	2.31 ± 0.19	2.21 ± 0.15	−0.7	−4.3%	
**Modified T-test (s)**						
	NTG	7.61 ± 0.55	7.38 ± 0.55	−0.6	−3.0%	Group: *p* = 0.33, η^2^_p_: 0.02 Time: *p* < 0.001, η^2^_p_: 0.37 Interaction: *p* < 0.001, η^2^_p_: 0.17
	CON	7.39 ± 0.57	7.32 ± 0.64	−0.2	−0.9%	
**Plank (s)**						
	NTG	86.60 ± 32.54	86.91 ± 31.22	0.0	0.4%	Group: *p* = 0.34, η^2^_p_: 0.01 Time: *p* = 0.56, η^2^_p_: 0.01 Interaction: *p* = 0.30, η^2^_p_: 0.02
	CON	97.85 ± 55.90	96.73 ± 52.75	−0.0	−1.1%	
**Vertical jump (cm)**						
	NTG	21.92 ± 4.51	24.60 ± 4.80	0.6	12.2%	Group: *p* = 0.01, η^2^_p_: 0.11 Time: *p* < 0.001, η^2^_p_: 0.38 Interaction: *p* = 0.04, η^2^_p_: 0.07
	CON	19.79 ± 4.12	21.10 ± 3.80	0.3	6.6%	
**Medicine ball throw (m)**						
	NTG	4.28 ± 0.71	4.52 ± 0.72	0.5	5.6%	Group: *p* = 0.03, η^2^_p_: 0.07 Time: *p* = 0.002, η^2^_p_: 0.14 Interaction: *p* = 0.4, η^2^_p_: 0.01
	CON	3.96 ± 0.75	4.10 ± 0.68	0.3	3.5%	

Note: pretest and posttest results are presented as mean ± SD. Abbreviations: NTG, neuromuscular training group; CON, control group.

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
