# Peer review of "Effects of Neuromuscular Training on Motor Competence and Physical Performance in Young Female Volleyball Players"

_ijerph, 2020, doi:10.3390/ijerph17051755_

Round 1

Reviewer 1 Report

Thank you for the opportunity to review this manuscript, which considers some interesting, applied issues.  Your study appears to be novel, but as submitted needs considerable work on the presentation. The authors showed an interesting point about the effects of neuromuscular training on motor competence and motor fitness ….. in young female volleyball players. Unfortunately, this investigation showed several hard points to overcome. There are several points when the terms “competence and motor fitness” was misleading used.

I suggest to the author to change the terms “competence and motor fitness” and choose a usual terminology (i.e. its usual physical fitness / motor ability but not motor fitness or motor competence)

Line 13: please to include mean/SD vs. ten to twelve.

Line 16: KTC acronym is unclear please to justify it

Line: 34: changes of direction vs. direction changes ……

Introduction.

Line 37/39: this two sentence are disconnected / please to update it

Line 68: please check body height in cm more than in m

For each test I suggest to include a reference (You can see a similar test https://www.ncbi.nlm.nih.gov/pubmed/31736783)

For medicine ball throw I suggest  https://www.ncbi.nlm.nih.gov/pubmed/25289710

Reviewer 2 Report

The authors raised an interesting problem of using strength training in children's. The impact of precisely described strength training (of 8-weeks duration) was assessed, using the KTK battery tests for motor competence assessment and other 5 test for motor fitness assessment in young female volleyball players. The results of presented work have application significance what is advantageous. The paper is well written, the text is consistent and logical. The minor flaw are listed below.

Methods

What dose abbreviations KTK and MQKTK means. The article will also be read by people who are not directly involved in shaping physical fitness. Not everyone knows what KTK is. Please, do not force the reader to search for the meaning of the abbreviation used. It may cause a decrease in interest of your work.

Line 98; “KTK form” I would rather change for KTK battery test

Line 121; what was the platform weight? If you precisely describe the bench sizes (3 previous paragraphs) and platform size why don't you put information about its weight? Was it 0,5 kg or 2kg, or…? The weight of the platform can affect the number of transfers.

Line 163; comment as above

Discussion

Line 286; “...decrease of neuromuscular control in females…” I would be careful with such a strong statement. It suggest as this state was of nervous origin which can be confusing. It is more about general change in girls posture with maturation what causes greater knee valgus and lower neuromuscular control comparing to boys.

Line 310; what INT means?

Reviewer 3 Report

General comments:

The purpose of this study was to examine the effects of neuromuscular training on performance in young volleyball players. Overall this was a well written and explained project. I appreciated the clarity of the methods and a clear and concise statistical analysis. The discussion has sensible and clear outcomes related to your results and refers to existing studies. I have mainly small comments that help with readability and clarification for the readers. I will include line by line comments below.

Line 16: KTK is abbreviated but not explained what this means.

Line 19: Unclear what MQKTK is. Please explain these for the readers

Line 23: motor fitness is a very broad term. May want to replace it with performance. This should be done throughout the whole manuscript

Line 33: suggested to rewrite this sentence to remove motor fitness and change to performance abilities

Line 40: We cannot prove anything in science outside of math. Please rewrite to say evidence supports

Line 92: Need to define KTK as this is the first place it is used

Line 99-100: do you mean reliability and validity has been shown with previous studies? Please rewrite this for clarity

Line 131: change motor fitness to performance test

Line 132: sprint 10 is bolded please edit with no bold and reword to “10m Sprint”

Table 2: does forward lounges and backward lounges need to be “lunges”

Table 2: need a semicolon after medicine instead of a period

Line 202-203: need to include in the statistical analysis section how you calculated ICC’s

Line 222: remove period between neuromuscular and training

Line 227: change motor fitness to performance tests

Line 228: change motor fitness to performance

Line 230: remove period between the training group and CON and change to a semicolon

Line 248: change motor fitness to performance tests

Line 264: need to define MQ before abbreviating

Line 307: should west be vest in both cases?

Line 307-309: this reads another study but no citation is provided

Line 342: check this citation to make sure the format is correct it says “undefined”

Line 350:  check this citation to make sure the format is correct it says “undefined”

Reviewer 4 Report

  1. Introduction L:27

His/Her conceptual vision of motor competence is reduced. Please, it must be contemplated in all its extension. Include the driving decision linked to the volleyball players in a motor interaction sport. I suggest that the authors include all the bibliography in introduction and will surely also be necessary in discussion. It should also be included as one of the great limitations of the study at the end of the work.

Ruiz, L. M., & Palomo, M. (2018). Clumsiness and Motor Competence in Physical Education and Sport Pedagogy. En O. Bernard & N. Llevot (Eds.), Pedagogy. Volume 2. IntechOpen, https://doi.org/10.5772/intechopen.70832

Ruiz, L.M.; Palomo, M.; Gómez, M. A., & Navia, J.A. (2018). When We Were Clumsy: Some Memories of Adults who were Low Skilled in Physical Education at School. Journal of Physical Education and Sports Management, 5(1), 30-36. https://doi.org/10.15640/jpesm.v5n1a4

Pic, M., & Lavega-Burgués, P. (2019). Estimating motor competence through motor games. RICYDE. Revista internacional de ciencias del deporte. 55(15), 5-19. https://doi.org/10.5232/ricyde2019.05501

Pic, M., Lavega-Burgués, P., & March-Llanes, J. (2019). Motor behaviour through traditional games. Educational Studies45(6), 742-755.

  1. Introduction L:62-65

Be more specific with the objectives of the study. The objective cannot be to investigate. I guess it will be determine or some similar verb, please modify this.

  1. Method

I miss the study design. This is very important, please, it must be included.

2.5 Data Analysis L:190

I do not find the test of normality of the data, please, it must be included.

  1. Discussion

- It should be included the approach of gender vision, specifically with relation of cooperation and motor opposition. This is an appropriate reference.

Miguel Pic, Vicente Navarro-Adelantado, Gudberg K. Jonsson. (2020) Gender Differences in Strategic Behavior in a Triadic Persecution Motor Game Identified Through an Observational Methodology. Frontiers in Psychology 11.

-The applications of the study must be developed in depth.
